# Prevalence of New and Established Avian Chlamydial Species in Humans and Their Psittacine Pet Birds in Belgium

**DOI:** 10.3390/microorganisms10091758

**Published:** 2022-08-31

**Authors:** Anne De Meyst, Rachid Aaziz, Joachim Pex, Lutgart Braeckman, Morag Livingstone, David Longbottom, Karine Laroucau, Daisy Vanrompay

**Affiliations:** 1Laboratory of Immunology and Animal Biotechnology, Department of Animal Sciences and Aquatic Ecology, Faculty of Bioscience Engineering, Ghent University, 9000 Ghent, Belgium; 2Bacterial Zoonoses Unit, Animal Health Laboratory, Anses, University Paris-Est, 94706 Maisons-Alfort, France; 3Department of Public Health, Faculty of Medicine and Health Sciences, Ghent University, 9000 Ghent, Belgium; 4Moredun Research Institute, Pentlands Science Park, Bush Loan, Edinburgh EH26 0PZ, UK

**Keywords:** animal chlamydiosis, *Chlamydia psittaci*, *Chlamydia avium*, zoonosis

## Abstract

The presence and zoonotic transfer of four different avian *Chlamydia* spp. was assessed in an epidemiological study in a psittacine bird population and its owners. Fecal swabs from 84 pet birds and pharyngeal swabs from 22 bird owners were collected from 21 locations in Flanders. Samples were examined using established and novel PCR platforms combined with culture on PCR-positive samples. *Chlamydiaceae* DNA was detected in 33 of 84 (39.3%) birds. The predominant part of the avian infections could be attributed to *C. psittaci* (22 of 84; 26.2%), followed by *C. avium* (11 of 84; 13.1%). *C. gallinacea* and *C. abortus* were not detected in birds or humans. *C. psittaci* was the only species detected in pet bird owners (4 of 22; 18.2%), stressing its zoonotic importance. This study showed that *C. psittaci* and the more recently discovered novel avian species *C. avium* are undoubtedly present in the Flemish psittacine bird population. Our results justify additional research in a larger psittacine bird population and its owners, focusing on *C. psittaci* and *C. avium*. In the meantime, increased awareness among pet bird owners and the implementation of preventive measures in the pet bird industry is advised to limit the circulation of established and novel emerging avian chlamydial species.

## 1. Introduction

*Chlamydiaceae* are Gram-negative obligate intracellular bacteria which replicate in mucosal epithelial cells. At present, the family comprises 14 characterized species which infect a variety of hosts, including companion animals and humans. Their wide host range is continuously increasing and new *Candidatus* species are regularly proposed. Due to their ability to cause severe illness, they pose a serious threat to both animal and human health [1].

One of the oldest recognized etiologies of *Chlamydia* respiratory disease in psittacine birds is *Chlamydia psittaci*. A *C. psittaci* infection may result in asymptomatic, acute or chronic illness. The infection causes pneumonia, air sacculitis, pericarditis and/or hepatitis and systemic disease can ultimately be lethal [2]. This airborne pathogen is highly prevalent in psittacine pet birds but reported prevalence rates vary depending on the diagnostic test used [3,4,5]. *C. psittaci* can be transmitted to humans, causing psittacosis, which is characterized by flu-like symptoms, atypical pneumonia and sometimes multi-organ failure [2]. Psittacosis is both underestimated and underdiagnosed as many clinicians, general practitioners, veterinarians and bird fanciers are not aware of its occurrence or its clinical manifestations [6]. According to Hogerwerf et al. (2017), around 1% of all community-acquired pneumonia cases are caused by *C. psittaci* [7]. 

*Chlamydia abortus*, in the pre-genomic era named *C. psittaci* serovar 1, infects small ruminants, causing ovine enzootic abortion [8]. While *C. abortus* is recognized as descended from *C. psittaci*, genomics also revealed the existence of intermediary strains between the two species that are recent evolutionary ancestors of *C. abortus* [1,9]. These *C. abortus* ancestors, variously referred to as atypical *C. psittaci*/*C. abortus*, *C. psittaci/C. abortus* intermediates or avian *C. abortus* strains, have been found in psittacine birds [3] and more recently in poultry [10]. Mammalian *C. abortus* strains can be transmitted to humans, mainly endangering the health of pregnant women, but the zoonotic potential of avian *C. abortus* strains remains to be investigated [8,10].

*Chlamydia avium* is a new *Chlamydia* species with zoonotic potential and was originally isolated in Italy from pigeons [11]. So far, the species has been predominantly encountered in asymptomatic infections, but recent reports provide proof of its virulence, as *C. avium* was identified in several outbreaks of respiratory disease in psittacine birds and pigeons [11,12].

*Chlamydia gallinacea* is also a new species in the genus *Chlamydia* [11]. The agent is currently widespread and emerging in chickens and has also been found in wild common Australian parrots (Crimson Rosella; *Platycercus elegans*) with a prevalence of 4.6% [5,13]. However, additional information on the epidemiology, zoonotic potential and pathogenicity of *C. gallinacea* for *Psittaciformes* is limited. Only recently, *C. gallinacea* was detected in human sputum samples from workers of *C. gallinacea*-positive poultry farms, suggesting animal-to-human transmission [14]. Pathogenicity studies in SPF chickens showed that *C. gallinacea* is less pathogenic compared to other *Chlamydia* species as only growth retardation was noticed [15,16]. 

Our current knowledge on old and emerging *Chlamydiae* highlights the need for further study and surveillance of these avian *Chlamydia* spp. in Psittaciformes. Additionally, interest in psittacine birds as pets is continuously increasing, whereby pet shop workers, veterinarians and pet bird owners are especially at risk of becoming infected with *Chlamydia* spp., as they regularly come into close contact with these pet birds [6]. In this study, samples from humans and their psittacine pet birds were collected to assess the presence and possible zoonotic transfer of established and new emerging *Chlamydia* spp.

## 2. Materials and Methods

### 2.1. Study Design and Sampling

Psittacine bird owners in Flanders were recruited through the Belgian Association of Parakeet and Parrot Enthusiasts from November 2020 until February 2021. Only bird owners with at least four psittacine birds were allowed to participate in this study. After informed consent, participants received packets by regular mail containing sampling instructions, eight aluminum rayon-tipped swabs for bird sampling (Copan, Brescia, Italy), two FlocQSwabs^®^ (Copan) for human sampling, DNA-RNA stabilization reagent (Merck Life Science BV, Overijse, Belgium) and standard Chlamydia transport medium. A medical questionnaire was also provided, informing about potential risk factors associated with a *Chlamydia* infection (e.g., the general health status of the bird owner, medication, (non)-professional activities, contact frequency with other animals, the general health status of the birds, bird medication and their housing). Volunteers were asked to take a swab from fresh fecal droppings originating from four different psittacine birds inside the aviary/birdcage in addition to a human pharyngeal swab (self-sampling). Each sample type was collected twice at the same time, depositing the first swab in DNA/RNA stabilization reagent for molecular testing and the second swab in transport medium for *Chlamydia* culture. Completed packages were sent back to the laboratory by express post. Upon arrival, swabs were shaken for 1 h at 4 °C, pseudonymized and stored at −80 °C (BSL3).

### 2.2. Molecular Diagnosis 

The presence of established and novel emerging *Chlamydia* species in human and avian samples was examined with different polymerase chain reaction (PCR) assays. Table 1 presents, for each assay, the target gene and primer and probe sequences. Genomic DNA was extracted with the QIAamp^®^ DNA mini kit (Qiagen, Antwerp, Belgium). 

First, the occurrence of *Chlamydiaceae* spp. was examined with a broad-range real-time PCR targeting the 23S rRNA gene of *Chlamydiaceae*. The PCR has a sensitivity of 3 Inclusion Forming Units (IFU) [17]. Samples with a Cycle threshold (Ct) value above 39 were considered negative. Second, *C. psittaci* was assessed in all samples using an *ompA*-based nested PCR which employs a SuperTaq polymerase enzyme (Cambio, Cambridge, UK) with enhanced sensitivity. In this PCR, all *C. psittaci* genotypes can be detected with a sensitivity of 1 IFU [18]. Nested-PCR-positive samples were subsequently genotyped with a novel combined PCR/high-resolution melting curve assay [19]. This assay is based on the analysis of eight phylogenetically informative Single Nucleotide Polymorphisms (SNPs) discriminating between eight different *C. psittaci* subtypes (I_Psittacine; II_Duck; III_Pigeon; IV_Turkey; V_Mat116; VI_M56; VII_VS225; VIII_WC). Third, *C. abortus*, *C. gallinacea* and *C. avium* DNA was pre-amplified from all samples for 15 cycles using the PerfeCTa PreAmp SuperMix (VWR, Fontenay-sous-Bois, France) and a mix of primers targeting each of the three species specifically. The pre-amplification was executed to be able to detect very low concentrations of DNA. Their presence was consequently confirmed by performing a *C. gallinacea*-specific *enoA* PCR assay [20], a *C. avium*-specific *enoA* PCR assay [21] and a *C. abortus*-specific *enoA* PCR assay, allowing the detection of the ruminant and avian strains of *C. abortus* [22]. All samples were tested in duplicate for each PCR. Only double-positive samples were judged as truly positive.

### 2.3. Culture

PCR-positive samples (41 samples in total) were analyzed for the presence of viable *Chlamydia* spp. in BSL3 facilities. Briefly, samples were inoculated onto Buffalo green monkey (BGM) cells as previously described [23] and incubated for six days at 37 °C. Afterwards, *Chlamydia* was visualized with the IMAGEN™ Chlamydia kit (Oxoid™, Geel, Belgium). Samples were considered positive when inclusion-forming units and/or mini-inclusions were present in the BGM cells.

### 2.4. Data Analysis

The questionnaires were used to determine potential risk factors associated with *Chlamydia* infections. Risk factors were determined by implementation of a generalized linear model with a binary logistic response. In order to determine whether each variable was significant to the model, a Wald Chi-Squared Test was used. Data analysis was performed in SPSS Statistics 27 with a significance level of 0.05.

## 3. Results

### 3.1. Descriptive Analysis 

Samples were collected from 22 bird owners and 84 psittacine birds from 21 locations across Flanders (Belgium). Three bird families were tested: cockatoos (*Cacatuidae*), new world parrots (*Psittacidae*) and old world parrots (*Psittaculidae*). All bird owners indicated that the birds appeared healthy and did not show any of the following signs at the moment of sampling: pumping breath, open-mouth breathing, abnormal breath sound, nasal or ocular discharge, ruffled feathers and/or diarrhea. Nine of twenty-two (40.9%) bird owners suffered from a cough, heavy breathing and/or a headache, clinical signs that may be due to psittacosis infection. Five of twenty-two (22.7%) bird owners reported that their birds had suffered from avian chlamydiosis in the past and in one of these cases the pathogen was even transmitted to the owner, who developed severe pneumonia. The patient immediately reported the possession of pet birds but was only diagnosed (clinical laboratory testing) and treated for psittacosis 7 days later. 

### 3.2. Prevalence of Chlamydiaceae

*Chlamydiaceae* DNA was detected in 33 of 84 (39.3%) birds and 2 of 22 (9.1%) bird owners from 13 of 21 (61.9%) locations. Excretion loads were rather low in both birds and humans, with Ct-values ranging from 31.9 to 39.0 (36.1 ± 2.3) and 38.4 to 38.8 (38.6 ± 0.3), respectively. Among the positive PCR samples, 15 of 33 (45.5%) bird samples and 1 of 2 (50%) human samples were positive in culture as well. An overview of the molecular test results can be found in Table 2.

### 3.3. Prevalence of Established and Novel Emerging Chlamydia spp.

*Chlamydia psittaci* DNA was present in 22 of 84 (26.2%) bird samples, originating from 9 of 21 (42.9%) different sampling locations. Four of twenty-two (18.2%) bird owners were PCR-positive. Three birds and three humans tested positive with the *C. psittaci*-specific nested PCR but were negative with the less sensitive *Chlamydiaceae*-specific PCR. From the *C. psittaci*-positive PCR samples, 12 of 22 (54.5%) bird samples and 3 of 4 (75%) human samples were also positive in culture. In total, genotyping revealed subtype I_Psittacines in 21 of 26 (80.8%) samples, which corresponds to strains belonging to the *C. psittaci* genotype A. The other *C. psittaci*-positive samples (including three positive human samples) could not be genotyped. 

*Chlamydia avium* DNA was found in 11 of 84 (13.1%) bird samples with Ct-values ranging from 35.8 to 40.5 (39.2 ± 1.2). Six of these eleven (54.5%) samples were also positive in culture. Seven birds were infected by both *C. psittaci* and *C. avium*. All human samples tested negative in the *C. avium*-specific PCR. 

*Chlamydia gallinacea* and *C. abortus* DNA could not be detected in any of the samples. 

In 11 *Chlamydiaceae*-positive samples (10 bird samples and 1 human sample), the species could not be identified. From those 11 samples, 3 bird samples were positive in culture as well.

A detailed overview of all individual test results can be found in Appendix A.

### 3.4. Risk Factor Determination

Contact with other bird orders (e.g., *Galliformes, Anseriformes, Columbiformes*), the origin of the birds (e.g., breeder facility, quarantine station, pet shop or other location), housing of the birds (e.g., exterior aviary, interior bird cage or both), earlier antibiotics treatments of the birds and the bird species were all considered as possible risk factors to the development of *Chlamydia* infections. None of the examined risk factors were shown to contribute significantly to the development of infections in birds or humans, but trends were noted. Fewer infections were observed when birds were predominantly housed in outdoor aviaries compared to birds that were housed both inside and outside or only inside. More infections were observed when more than two different bird species were housed at the same location. Furthermore, the four bird owners that were PCR-*C. psittaci*-positive indicated that they had daily contact with their pet birds. The only bird owner that was positive in PCR but negative in culture indicated that the birds were housed in outdoor aviaries alone.

## 4. Discussion

Samples from humans and their psittacine pet birds were examined for the presence of established and novel emerging avian *Chlamydia* spp. to gain better insight into their prevalence and zoonotic potential in a risk population. One family-specific and four species-specific PCRs were executed, combined with culture on 84 fecal bird swabs and 22 human pharyngeal swabs from 21 different locations. 

*Chlamydiaceae* were found in 39.3% of the birds and 9.1% of the bird owners. Excretion loads were low, resulting in rather high Ct values with the family-specific PCR and may be explained by ground fecal sampling rather than pharyngeal or cloacal swab analysis on birds. We opted for the former as these pet birds are naturally very sensitive creatures and tend to be unable to deal with stress. Additionally, taking fresh fecal (floor) samples required no specific sampling expertise on the part of the bird owners. However, fecal chlamydial excretion in birds is intermittent [9,24], meaning that the results are an underestimate of the true current number of positive birds. 

Most chlamydial infections were attributed to *C. psittaci*, followed by *C. avium*. *C. gallinacea* and *C. abortus* were not found in this study. The prevalence rates found in this study were high and indicate that the novel *C. avium* species and especially *C. psittaci* are present in psittacine birds in Flanders. However, this pilot study is limited as it only has a small sample size and only incorporated bird owners with at least four psittacine birds, resulting in a selection bias. More birds and especially single-bird owners should be sampled to be able to generalize these prevalence rates to the wider population.

Overall, *C. psittaci* remains the number one *Chlamydia* species in psittacine pet birds in Flanders. Its prevalence was investigated earlier in 2009 when samples were collected from 39 Belgian breeding facilities and 19.2% of the birds and 13.0% of the bird owners were found to be *C. psittaci*-positive [4]. In the present study, the prevalence of *C. psittaci* was even higher (26.2% in birds and 18.2% in humans). This indicates that the current measures to avoid *C. psittaci* transmission to psittacine birds and their owners are not sufficient and more care should be taken to control infections. Transmission of *Chlamydia* spp. in pet birds in Europe is restricted by directives/guidelines at both a European and national level. According to European directive EC 318/2007 [25], psittacine birds imported from third countries must stay in a designated licensed quarantine station for at least 21 days and must undergo two veterinary inspections. Only in cases of a suspected or confirmed *C. psittaci* infection must all birds be treated accordingly. A limitation of this directive is that the birds do not have to be tested for the presence of *Chlamydia* spp. In this way, (asymptomatic) infections can easily be missed. On a national level, guidelines are available to prevent contact between infected birds and humans. Examples are the mandatory notification after a confirmed psittacosis case and a contact investigation and source detection when necessary. Bird monitoring and the use of health and/or vaccination certificates in pet shops or at breeder facilities would further reduce the spread of these agents but is not mandatory [26]. These limited/incomplete measures contribute to the continuous circulation of *Chlamydia* spp. in Psittacines and its transmission to humans.

In 20 out of 22 *C. psittaci*-positive bird samples, the strain could be typed as genotype A. This is the dominant genotype in psittacine birds, known to be highly virulent and able to cause severe avian chlamydiosis and psittacosis in humans. Other genotypes like Mat116, CPX0308 and genotype F have been detected in psittacine birds but only occasionally [2,6]. One *C. psittaci*-positive human sample could be typed as genotype A as well, illustrating the transmission of this genotype from infected birds to the bird owner. The three other pharyngeal *C. psittaci*-positive human samples could not be genotyped, possibly due to low DNA concentrations in the sample. 

Considering the recent discovery of *C. avium*, the available epidemiological data is rather limited. The prevalence of this pathogen in feral pigeons was investigated in the Netherlands by PCR and was found to be remarkably high (23 of 81 birds; 28.4%) [27]. In psittacine birds, the agent has only been reported occasionally [11,28,29]. To our knowledge, this is the first study where *C. avium* is detected during a pilot epidemiological study in psittacine birds with a significant incidence rate (13.1%). The zoonotic potential of the agent could not yet be confirmed, as none of the human samples were found to be *C. avium*-positive.

*C. gallinacea* and *C. abortus* were not identified in any of the samples in this study but their presence in pet birds would not be unusual. On the other hand, avian *C. abortus* strains have mainly been detected in wild birds and *C. gallinacea* is more related to poultry [9,13]. It still remains to be discovered whether these two species are actively circulating in companion birds or if they only occasionally cross their traditional host barrier [3,5].

Interestingly, several samples were positive in a broad-range *Chlamydiaceae*-specific PCR but negative for *C. psittaci*, *C. abortus*, *C. avium* and *C. gallinacea*. The existence of other chlamydial species in psittacine birds has not yet been confirmed but is credible as *C. trachomatis* and *C. pecorum* have already been detected in urban pigeons [30] and Guo et al. (2016) reported the occasional presence of *C. pecorum*, *C. suis* and *C. muridarum* in domestic birds [16]. Unfortunately, further characterization of these strains was hampered as attempts to cultivate the strains were unsuccessful. 

Although *Chlamydia* infections can result in severe illness, all the birds were, according to the owner, healthy at the time of sampling. However, the clinical condition of the birds was not examined by experienced veterinarians and birds have also been known to be professionals at hiding illness to protect themselves against predators. As a result, clinical signs of a *Chlamydia* infection could have been missed. On the other hand, asymptomatic persistent chlamydial infections, with only mild or even no bacterial shedding, do occur in Psittacines. Asymptomatic birds still act as carriers and can therefore transmit the agent to other, more sensitive birds or to humans. When the bird’s immune system is suppressed (e.g., during stress or concurrent infections), chlamydial replication can be re-initiated, resulting in severe respiratory disease and renewed excretion [2]. 

In this study, *C. psittaci* was transmitted to four bird owners. Three of them reported experience of clinical signs that could be related to psittacosis (coughing, dry throat, tight feeling on the chest, rhinitis) but these were not clinically evaluated. Furthermore, all samples were collected during the respiratory season (winter 2020) and an early phase in the COVID-19 pandemic when no vaccine was yet available and, therefore, other pathogens might have caused these symptoms as well. Human *C. psittaci* infections are easily missed due to their flu-like course and are therefore largely underdiagnosed. Hence, more testing is recommended, for example, by including *C. psittaci* in the routine diagnostic panel for human community acquired pneumonia [6]. 

From the investigated risk factors, none appeared to be significantly contributing to the development of *Chlamydia* infections. A larger study population is recommended in future studies for conducting a deeper investigation into such risk factors. However, when analyzing trends in our dataset, daily contact with birds, indoor housing of birds and housing of multiple bird species in the same location all seemed to contribute to higher incidence rates. The frequency of exposure to birds and the bird species have indeed been proven to be crucial risk factors. Cockatiels and budgerigars are especially considered to be more susceptible but this could not be confirmed in this study [31].

This study indicates that psittacine birds are an important reservoir for *Chlamydia* spp. and more care should be taken to limit the spread of these bacteria. Balsamo et al. (2017) provided a thorough overview of preventive measures, including quarantining newly purchased birds and following strict cleaning and disinfection measures. Vaccination is a putative preventive measure and, although 82% of the bird owners indicated in the questionnaire that they would be willing to vaccinate their birds, no commercial vaccine against *Chlamydia* spp. in pet birds is available yet. Once an infection does occur, birds should be treated with antibiotics as fast as possible. For complete eradication of *Chlamydia*, a treatment period of 30–45 days is recommended in pet birds [31]. Remarkably, pet bird owners indicated that they have treated their birds before, but never longer than 14 days. Additionally, in a former, similar study, some pet bird owners admitted to the prophylactic use of antibiotics, bought on the internet off-label, as a preventative treatment (before the breeding season) [4]. Reduced treatment periods increase the risk of treatment failure and are dangerous for both animal and human health as the zoonotic pathogen is not completely eradicated. Additionally, new species may arise in this way as horizontal gene transfer, antibiotic pressure and homologous recombination are able to shape the evolution of *Chlamydia* [32]. Therefore, we opt for better communication between the government, researchers, veterinarians, medical physicians and people working in the pet bird industry in order to prevent further spread of these species and efficiently control infections upon diagnosis.

## Figures and Tables

**Table 1 microorganisms-10-01758-t001:** Target gene and primer and probe sequences for the different PCR assays.

PCR	Target Gene	Primer and Probe Sequences	References
*Chlamydiaceae*	*23S rRNA*	Fw: 5′-CTGAAACCAGTAGCTTATAAGCGGT-3′ Rv: 5′-ACCTCGCCGTTTAACTTAACTCC-3′ Pr: 5′-FAM-CTCATCATGCAAAAGGCACGCCG-TAMRA-3′	[17]
*C. psittaci*	*ompA*	Fw_extern: 5′-CCTGTAGGGAACCCAGCTGAA-3′ Rv_extern: 5′-GGTTGAGCAATGCGGATAGTAT-3′ Fw_intern: 5′-GCAGGATACTACGGAGA-3′ Rv_intern: 5′-GGAACTCAGCTCCTAAAG-3′	[18]
*C. gallinacea*	*enoA*	Fw: 5′-CAATGGCCTACAATTCCAAGAGT-3′ Rv: 5′-CATGCGTACAGCTTCCGTAAAC-3′ Pr: 5′-FAM-ATTCGCCCTACGGGAGCCCCTT -TAMRA-3′	[20]
*C. avium*	*enoA*	Fw: 5‘-CATGCAAGCTATTGAGAAAAGTGGT-3′ Rv 5′-CCTTGATATGTACGTGTTTTCTCG-3′ Pr: 5′-FAM-CACCCCTGGTGAAGATATTTCCTTAGCAT-TAMRA-3′	[21]
*C. abortus*	*enoA*	Fw: 5‘-AACAACGGCCTGCAATTTCAA-3′ Rv 5′-TGAGAAGGTTTTTCAATGTATGGAAC-3′ Pr: 5′-FAM-GGCACCCATACGTACAGCTTCTTG-BHQ1-3′	[22]

**Table 2 microorganisms-10-01758-t002:** *Chlamydia* infections in psittacine birds and their owners as diagnosed by PCR, Belgium, 2020–2021.

	Prevalence in Birds	Prevalence in Humans	Positive Locations
*Chlamydiaceae*	33/84 (39.3%)	2/22 (9.1%)	13/21 (61.9%)
*C. psittaci*	22/84 (26.2%)	4/22 (18.2%)	9/21 (42.9%)
*C. avium*	11/84 (13.1%)	0/22 (0.0%)	9/21 (42.9%)
*C. gallinacea*	0/84 (0.0%)	0/22 (0.0%)	0/21 (0.0%)
*C. abortus*	0/84 (0.0%)	0/22 (0.0%)	0/21 (0.0%)

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
