# Peer review of "Prevalence of New and Established Avian Chlamydial Species in Humans and Their Psittacine Pet Birds in Belgium"

_microorganisms, 2022, doi:10.3390/microorganisms10091758_

Round 1
Reviewer 1 Report
Authors' response to reviewer 3
1. Comment on line 172: The table indicates that 4 people are C. psittaci-positive but only 2 people are Chlamydiaceae-positive. Indeed, and this can be explained by the fact that the C. psittaci-specific nested PCR (1 IFU) is more sensitive than the Chlamydiaceae-specific qPCR (3 IFU). This is mentioned in line 162-163 of the revised paper. Further, the nested PCR protocol uses a super taq polymerase which is a mix of three different enzymes. This mix of enzymes has been proven to be very effective in the past, even in the presence of inhibitors in bird faeces and human pharyngeal samples (unpublished data).
Concerning mentions by authors on the use of a super Taq polymerase which is a mix of three different enzymes. In the material and methods of this manuscript, they report the use of the KIT PerfeCTa PreAmp SuperMix (VWR, Fontenay-sous-Bois, France). This kit is an ultra-pure, highly processive, thermostable DNA polymerase combined with high avidity monoclonal antibodies. These proprieties in the polymerase mix made that ​​resistant to PCR inhibitors. They provided a stringent automatic hot-start that allows reaction assembly and temporary storage at room temperature before pre-amplification, according to the manual description of the kit.
On the other hand, the nested PCR protocol used in this manuscript is cited in reference number 18 (Van Loock M, Verminnen K, Messmer TO, Volckaert G, Goddeeris BM, Vanrompay D. Use of a nested PCR-enzyme immunoassay with an internal control to detect Chlamydophila psittaci in turkeys. BMC Infect Dis 2005;5:1–9); they describe the use of super-Taq polymerase; however, they do not describe either the brand or the company. The looking for this product on the web, I found super-Taq polymerase from ABclonal technology (www.abclonal.com.cn). The supplier's instructions do not mention using three different enzymes, as the authors say. Could you note which Taq polymerase was used in your manuscript and a description from it with the company name? This response is critical if any person wants the reproduction of these PCR assays.
Author Response
Dear reviewer,
We added additional clarification on the used enzyme in the revised manuscript (line 109). The SuperTaq enzyme we used was produced by Cambio but we did notice that the composition of the enzyme mix is no longer available on the website. Therefore, we were not able to provide additional information.
Kind regards,

Reviewer 2 Report
All questions were answered by the authors. I still think it would be interesting to examine such well defined samples for the presence of novel avian chlamydia, even if it just confirms their host specificity.
All suggestions were incorporated in the manuscript.
Author Response
Dear reviewer,
Thank you for the comments. We agree that testing for Chlamydiifrater (or other new avian Chlamydia) can be useful in the future.
Kind regards,